# The Ecology and Evolution of Japanese Encephalitis Virus

**DOI:** 10.3390/pathogens10121534

**Published:** 2021-11-24

**Authors:** Peter Mulvey, Veasna Duong, Sebastien Boyer, Graham Burgess, David T. Williams, Philippe Dussart, Paul F. Horwood

**Affiliations:** 1Australian Institute of Tropical Health and Medicine, James Cook University, Townsville 4811, Australia; peter.mulvey@daf.qld.gov.au; 2Institut Pasteur du Cambodge, Institut Pasteur International Network, Phnom Penh 12201, Cambodia; dveasna@pasteur-kh.org (V.D.); sboyer@pasteur-kh.org (S.B.); pdussart@pasteur.mg (P.D.); 3College of Public Health, Medical and Veterinary Sciences, James Cook University, Townsville 4811, Australia; graham.burgess@jcu.edu.au; 4Australian Centre for Disease Preparedness, Commonwealth Scientific and Industrial Research Organisation (CSIRO), Geelong 3220, Australia; D.Williams@csiro.au

**Keywords:** Japanese encephalitis, JEV, *Culex*, pigs, wading birds, ecology, epidemiology, flavivirus, Asia, zoonotic

## Abstract

Japanese encephalitis virus (JEV) is a mosquito-borne flavivirus mainly spread by *Culex* mosquitoes that currently has a geographic distribution across most of Southeast Asia and the Western Pacific. Infection with JEV can cause Japanese encephalitis (JE), a severe disease with a high mortality rate, which also results in ongoing sequalae in many survivors. The natural reservoir of JEV is ardeid wading birds, such as egrets and herons, but pigs commonly play an important role as an amplifying host during outbreaks in human populations. Other domestic animals and wildlife have been detected as hosts for JEV, but their role in the ecology and epidemiology of JEV is uncertain. Safe and effective JEV vaccines are available, but unfortunately, their use remains low in most endemic countries where they are most needed. Increased surveillance and diagnosis of JE is required as climate change and social disruption are likely to facilitate further geographical expansion of *Culex* vectors and JE risk areas.

## 1. Introduction

Japanese encephalitis virus (JEV) is one of the most important causes of human viral encephalitis in Asia [1]. JEV is a zoonotic, vector-borne virus, spread primarily by *Culex* mosquitoes. Various species of birds are the natural reservoir, while pigs are considered the main maintenance or amplifying host. JEV infection in humans generally leads to mild febrile illness, while approximately 1% of infected people develop Japanese encephalitis (JE), which has a mortality rate approaching 30%. The disease also has a high rate of neurological sequelae, with 25–30% of survivors developing lifelong disabilities or cognitive impairments [2]. The World Health Organization (WHO) estimates global JE cases at 68,000 people annually [3]. However, these figures are widely regarded as an underestimate, with actual disease burden probably closer to 175,000 cases annually [4,5]. 

Humans are a dead-end host for JEV infection, because viraemia is insufficient to be infectious to the mosquito vector [6]. JEV circulates in a natural cycle involving a range of animal species including *Culex* mosquitoes, wild wading birds, and pigs [7,8,9] (Figure 1). While birds are the natural reservoir for JEV, pigs act as amplifying hosts and are often associated with outbreaks amongst humans [10]. However, outbreaks do occur in the absence of pigs, which suggests that birds or other species may act as amplifying hosts in some instances [11]. JEV immunisation and eradication programs can practically eliminate human disease as evidenced in economically affluent Asian countries such as Japan, Korea, and Taiwan [12]. However, the geographical spread of JEV has continued to increase in Southeast Asia despite vaccination campaigns in some countries. The increasing spread of mosquito vectors, compounded by poor knowledge on JEV transmission due to a lack of surveillance [13,14,15], constitutes a threat for the further expansion of the geographical range of JEV and greater impacts to human populations. Global climate change also presents an opportunity for increased distribution of mosquito vectors and JEV into previously non-endemic areas as mosquito habitats expand.

## 2. Classification, Virion and Genome Structure

JEV is a flavivirus (genus Flavivirus, family Flaviviridae) with a single stranded, non-segmented, positive sense RNA genome. The JEV genome is approximately 11 kilobases in length and encodes a single open reading frame, which is translated into a large polyprotein that is proteolytically cleaved to form three structural proteins and seven non-structural proteins. Virions are approximately 50 nm in diameter and are spherical in shape. JEV belongs to the Japanese encephalitis serocomplex, along with other important arboviruses with which it shares a close genetic and antigenic relationship, such as West Nile virus (WNV), St. Louis encephalitis virus (SLEV), and Murray Valley encephalitis virus (MVEV) [16,17].

## 3. Epidemiology of Japanese Encephalitis

The JEV burden throughout Asia decreased during the late 1960s following aggressive pesticide use and the introduction of JEV vaccines in economically affluent countries [12,18]. However, increases in population growth, pork production, and irrigated rice agriculture throughout Southeast Asia has led to further spread of JEV and increased the burden of JE in recent decades [15,19,20]. In a study conducted in Vietnam, JEV was reported to cause 217 out of 421 (52%) acute meningoencephalitis (AME) cases from 1998–2007 [21]. Continued surveillance in Vietnam from 2007–2010 confirmed that 23% of AME cases in children (<14 years) were caused by JEV [22]. In Cambodia, JEV was reported as the primary cause of AME in children (<15 years), with 35% of 1160 patients confirmed or highly probable to have JEV infection [23]. These reports highlight JEV as a major causative agent for AME cases in children in Asia.

JEV outbreaks have also been reported in regions previously thought to be free of JEV, such as the Torres Strait region of Australia and Papua New Guinea [24,25]. In the Torres Strait, a JEV outbreak in the mid-1990s led to the removal of pig populations from human habitats and a widespread vaccination campaign [26,27]. Although relocation of pigs did not eliminate the risk of JEV circulation [28], there have been no additional human JE cases reported in the Torres Straits. However, ongoing mosquito and sentinel animal surveillance have demonstrated that the virus has continued to circulate in this region [29,30]. More recently, the first case of JE on the Tiwi Islands in the Northern Territory of Australia was reported in 2021 [31], raising concerns over future outbreaks in this region. Moreover, the detection of an autochthonous case of JEV infection in Angola in 2016 [32] raises the troubling possibility that the virus could be emergent in the African continent. 

Despite the high JE disease burden, surveillance is still lacking throughout most of Southeast Asia. Many countries lack the diagnostic capabilities to confirm JEV infection, and JE is often misdiagnosed based on symptoms similar to other viral infections. Thus, the status of JEV in the Asia-Pacific region is hard to assess. The virus is clearly endemic throughout the region, with high seroprevalence reported in wild and domestic animals and humans [33]. Health information is lacking in many countries that do not have the diagnostic capabilities to survey JEV burden. Clearly, transmission and infection rates of JEV are underestimated throughout the Asia-Pacific region. Increased surveillance and reporting of JEV infection need to be undertaken to realise the true burden of JE.

## 4. Clinical Picture

JE primarily affects children (<14 years) in endemic areas, while adults are also at risk in areas where epidemics occur [1]. Most JEV infections in humans are asymptomatic or cause mild flu-like symptoms that last for 5–15 days [34,35]. However, some infections manifest as encephalitis characterised by headaches, fever, seizures, and abnormal behaviour 2–4 days following infection. Haemorrhagic lesions develop in the brain, and the meninges become inflamed, characterised by neck stiffness. Paralysis, especially in the upper limbs, may follow. Mortality rates of JE can be as high as 30%, and those that recover may have lifelong mental impairment and physical ailments such as difficulties speaking and moving [34,36,37]. JEV remains the most common cause of childhood AME throughout Asia and therefore causes a significant morbidity and economic burden in affected countries [22,23,38,39].

## 5. Genetic Evolution of JEV

Although JEV was first detected and recognised as a public health threat in Japan, molecular studies suggest that ancestral JEV emerged from the Indonesia/Malaysia region and subsequently spread north [40]. JEV currently exists as five distinct genotypes, GI (clades GI-a and GI-b), GII, GIII, GIV, and GV [41] (Figure 2). Each genotype can be distinguished based on nucleotide sequence in the E-protein gene. However, all JEV strains belong to a single serotype, because neutralising antibodies are cross-reactive between genotypes [16].

GIII was the source of numerous JE outbreaks and was the most frequently isolated genotype throughout Asia until the 1990s [42,43]. Initially isolated in Cambodia in 1967, GI replaced GIII in the early to mid-1990s, and GI has since become the most frequent cause of JE outbreaks in the region [41]. GI has been isolated during recent outbreaks in Japan [44], Korea [45], India [46], China [47], Taiwan [48], Vietnam [49], Thailand [50], Malaysia [51], and Cambodia [52]. In the Australasian region, although the first outbreak in 1995 was caused by a GII strain, since 2000, only GI viruses have been detected [29,53]. Phylogeographic and phylodynamic analyses suggest that GI-a originated in Thailand in the 1940s and GI-b in Vietnam in 1950s. GI-b had seeded most of temperate Asia by 1983 before genetic diversity increased, until the 2000s, when the genotype became dominant in JEV endemic and naïve regions of Asia [54]. 

In an attempt to identify genetic determinants driving GI-b/GIII lineage replacement, GI-b and GIII genome and E protein sequences from Asian strains were analysed in silico to investigate amino acid mutations, positive selection, and host range [42]. Both lineages had low ratios of non-synonymous to synonymous substitutions (dN/dS), with a small number of individual sites under positive selection [42]. Historically, identification of positive selection in the genome of an emerging lineage would suggest a selective advantage. Surprisingly, GIII was shown to have a greater selective advantage, because it was predicted to be shaped by positive selection, while GI was predicted to be neutral [42]. The lower diversity of GI strains was associated with a narrower vector range than GIII strains, suggesting enhanced transmission or more efficient replication cycle between *Culex* sp. and pigs. This theory was supported by in vitro experiments that observed GI isolates with significantly higher infectivity titres in mosquito cell lines (C6/36) compared to GIII isolates [55]. GI isolates had more than an 11-fold increase in titres compared to GIII isolates, which suggested GI had better replicative fitness in mosquitoes and therefore represent a greater risk of transmission [55]. In another study, oral challenge of *Cx. quinquefasciatus* mosquitoes with either GI-b or GIII isolates resulted in no significant differences in virus ingestion, dissemination, or transmission rates in these mosquitoes [54]. However, a higher infection rate was found for the GIII isolate tested compared to the GI isolates [54], further confounding interpretations of the basis for the dominance of GI. Therefore, the biological mechanism underlying the GI replacement of GIII throughout Asia is not yet clear. Future in vivo challenge studies with the primary vector in Southeast Asia, *Cx. tritaeniorhynchus*, may be revealing.

## 6. JEV and Asian Economic Development

Asia has had unprecedented population growth, increasing from 2.1 billion in 1950 to 6.6 billion in 2010 [56]. To support this population growth, Asia increased agricultural yield through expansion of rice and domestic swine production. From 1963 to 2003, the total rice harvest area in JEV endemic countries increased 22%, from 1,102,459 km^2^ to 1,345,000 km^2^, which increased rice yield 134%, from 226 million tons to 529 million tons annually [57]. Currently, Asia accounts for approximately 91% of global rice production [20]. Swine are typically reared in open pens close to rural housing. From 1990 to 2005, pork production increased between 30–381% in JEV endemic countries [58]. These agricultural practices, combined with unprecedented population growth, may have contributed to JEV genotype replacement throughout Asia and the increasing burden of JE. For example, increased vector mosquito habitats and pig populations may have provided an ecological advantage for GI strains, for which there is evidence of optimised transmission between *Cx. tritaeniorhynchus* and pigs [42], compared to GIII viruses, which appear to have a more diverse host-vector range.

## 7. The Ecology of JEV

JEV is maintained in a natural transmission cycle amongst mosquitoes and wading birds, while pigs act as an amplifying host (Figure 1). Other domestic animals (cows, dogs, chickens, goats, and horses) and wildlife (flying foxes, frogs, snakes, and ducks) have been identified as dead-end hosts for JEV due to the low viraemia produced in these hosts, which is insufficient to infect the mosquito vector [33].

Host cell tropism is likely determined by JEV attachment and entry into host cells. Initial interaction of JEV to the host cell is thought to be through non-specific attachment of the virion, followed by highly specific binding of the envelope (E) protein to an unknown receptor [17]. Although much is understood about the viral processes involved in viral attachment and entry, little is known about the cellular aspects of this important process. JEV is able to infect and replicate in a range of cell types from different animal species, including mammals, birds and insects. As such, there are probably numerous cell factors involved in viral attachment and entry. Further elucidation of the viral–host interactions is an important area of research for possible therapeutics against JEV infection [17].

### 7.1. Mosquitoes

At least 14 mosquito species have been confirmed as JEV vectors, and experimental vector competence has been demonstrated in a further 11 species (reviewed in [59]). However, the major vectors of the virus are from the *Culex vishnui* subgroup, *particularly Cx. tritaeniorhynchus* [60]. Other *Culex* species including *Cx. bitaeniorhynchus*, *Cx. fuscocephala*, *Cx. gelidus*, *Cx. annulirostris*, and *Cx. quinquefasciatus* are important secondary JEV vectors that may also be primary vectors in some regions [59,61]. 

The distribution of *Cx. tritaeniorhynchus* extends from Pakistan in the West to Japan in the East, and from China/Korea in the North to Indonesia in the South [62]. It has also been found in Greece and Turkey [63,64], suggesting a possible continuum from Pakistan to Turkey via an unconfirmed presence in Iran. Recent detection of *Cx. tritaeniorhynchus* in northern Australia, an area with abundant wading birds and feral pigs, increases the risk for JEV establishment in this area [65]. *Cx. bitaeniorhynchus* is present in Africa, Asia, Southeast Asia, and Oceania [59]. *Cx. fuscocephala* is mainly present in Southeast Asia, and distributed from Pakistan (West), Timor (South), and Japan/China (Northeast) [59]. *Cx. gelidus* has almost the same distribution area as *Cx. fuscocephala*, but this extends to northern Australia [66]. *Cx. annulirostris* is found throughout the Australasia region and north to Indonesia/Malaysia, and it has been associated with previous outbreaks in the Torres Strait region of Australia [24]. Finally, *Cx. quinquefasciatus* has a cosmopolitan distribution with a preference for human habitats, including peri-urban and urban areas [67]. The distribution of these vector species closely correlates to the distribution of JEV (Figure 2), showing the adaptability of JEV to increase its distribution area and a strong probability of extension towards the East.

These six species have conquered many different larval niches, including temporary, semi-permanent, and permanent ground water habitats. These habitats include pools, puddles, small streams, rice fields, and human water containers. Irrigated rice fields provide breeding grounds for *Culex* sp. and can also attract migratory wading birds contributing to the maintenance of the natural transmission cycle [12]. *Cx. tritaeniorhynchus* larval habitats may also be found in wells, ponds, and ditches, in addition to storage containers within urban environments. These urban environments provide access to human blood meals and increase the potential for JEV transmission.

Blood meals isolated from *Cx. tritaeniorhynchus*, the major vector of JEV in Asia, indicate a trophic preference for cows over pigs [68]. Wild-collected mosquitoes fed on cows almost 10-fold more frequently (39–45.3%) than on pigs (2.4–5.3%) when both animals were present [69]. Even when only one animal was present (a single cow or pig), *Cx. tritaeniorhynchus* mosquitoes displayed a preference for cows (65.2–66.1%) over pigs (42.4–56.6%) [69]. More recently, despite an apparent preference of *Cx. vishnui*, *Cx. gelidus*, and *Cx. tritaeniorhynchus* for cows, modelling showed a preference of these mosquito species to pigs [70]. However, feeding preferences in the wild are dependent on host abundance and time of feeding. As an example, dog blood was detected in four main *Culex* species in Cambodia [70]. In Australasia, where *Cx. sitiens* subgroup mosquitoes, in particular *Cx. annulirostris*, are the major vector species of JEV, host preferences for bovines, dogs, pigs [71], and marsupials [72] have been reported. As such, the blood meals of the six major JEV vector (*Culex*) species will likely reflect the local domestic and wildlife populations where the mosquitoes are found, as they are known to be opportunistic and generalist feeders. 

### 7.2. Birds

Wild ardeid waterbirds (egrets and herons) have been implicated as the main wildlife reservoir for JEV and are recognised as important transmission hosts. A variety of other bird species are susceptible to JEV infection but are considered to play a minor role as viral reservoirs compared to egrets, herons, and pigs. JEV infection of ardeid birds is asymptomatic and leads to seroconversion [9,73]. Experimental infection has shown that egrets and herons with high viraemia are capable of transmitting JEV to mosquitoes [9]. The role of ardeid birds in JEV transmission has been supported by surveillance studies in villages near rice fields either with or without herons. Where heronries were absent, JEV neutralising antibody seroconversion in children 0–5 and 6–15 years age groups was 0–13% and 5–12%, respectively; whereas with herons, present seroconversion was 50% and 56%, respectively [74]. The seasonal migratory movement of ardeid birds encompasses all continents (except Antarctica) with high population movement in south, east, central, and north Asia. Thus, the migratory movement of ardeid birds potentially enhances JEV range and contributes to the transmission of the virus.

Domesticated birds, adult ducks, and chickens are thought to play only minor roles in JEV ecology, because they develop low viraemia that is unlikely to result in transmission to feeding mosquitoes [75]. However, recent studies report that JEV viraemia is significantly higher in juvenile ducklings and chicks, which may lead to transmission to feeding mosquitoes [76]. Age-related viraemia was shown in ducklings and chicks experimentally infected with JEV between 2 and 42 days of age [76]. Peak viraemia was > 105 pfu/mL in newborn hatchlings [76], which is a level that was previously shown in experimentally infected chicks to successfully transmit to feeding *Cx. tritaeniorhynchus* mosquitoes [8]. The authors noted JEV infection of chicks and ducklings was either subclinical or resulted in clinical signs similar to avian influenza infection, which may lead to outbreaks in poultry being mistakenly attributed to influenza or other poultry pathogens [76]. Considering that domesticated birds are housed in close proximity to humans throughout Asia, and live bird markets are commonly situated in densely populated cities, further field studies are needed to investigate the role of poultry in JEV transmission dynamics.

### 7.3. Pigs

Pigs serve as the principal amplifying hosts to JEV in epidemic areas and are maintenance hosts in endemic areas [7,10,77]. Concentrated pig farming in some regions of East and Southeast Asia results in pigs being the primary component of the domestic JEV transmission cycle [78,79]. Industrialisation of pig farming in Southeast Asia (Vietnam, Thailand and Myanmar), with enhanced amplification of JEV within dense pig herds, has contributed to an increased risk of JEV transmission [80]. Therefore, the vicinity of pig populations is the main risk factor for JEV transmission into the human population.

JEV infection of pigs generally remains asymptomatic or causes mild disease (reviewed in [81]), which means that outbreaks in swine may go unnoticed [7,82]. Immunologically naïve female sows may have increased stillbirth and abortion rates [83]. However, sows exposed to JEV before pregnancy tend to have lower pregnancy abnormalities [84]. Infection of boars has also been associated with infertility [85]. Encephalitis has also been observed in piglets following experimental infection [81,86].

In endemic areas, JEV seroprevalence in pigs is typically high (98–100%), and they develop high viraemia for 3–5 days following primary infection, which is sufficient for infecting biting *Culex* sp. mosquitoes [82,87]. Infected *Culex* sp. mosquitoes can transmit JEV to immunological naïve pigs, hence completing the transmission cycle [87]. Early evidence suggested that there are two viral amplification cycles in pigs. The initial cycle starts when infected mosquitoes bite pigs, leading to a ~20% infection rate. The second cycle involves infection of mosquitoes that have fed on viraemic pigs. These mosquitoes then transmit JEV to other naïve pigs, leading to up to 100% seroconversion amongst the pig population [7,88]. 

Recent studies have confirmed vector-free transmission of JEV between pigs via direct contact with infected animals through the oronasal route [89]. JEV RNA and live virus persisted in pig tonsils for six weeks, which was likely the primary site of replication and transmission [89]. Vector-free transmission is a possible explanation for the rapid seroconversion of newborn piglets to JEV [85]. In a Cambodian study, seroconversion of piglets began within one month of weaning [90]. Amongst 29 piglets, 28 had seroconverted before the age of six months (96.6%), which indicated intense transmission of the virus [90]. Despite the intensive circulation of JEV detected in pigs during this study, only one pool of *Cx. tritaeniorhynchus* collected over the same time tested positive for JEV [90]. The low prevalence of JEV positive mosquitoes in this study and others [91] suggests that vector-free transmission in pig populations could be an underappreciated factor in the epidemiology of JEV outbreaks.

JEV infection of pigs and the subsequent risk of outbreaks in humans can be reduced through vaccination of pig populations. Typically, JEV immunisation induces strong systemic IgM and IgG responses, which protect pigs against injected JEV challenge, with a resulting absence of transmission to feeding mosquitoes [92]. However, limitations to swine vaccination have led to low vaccination levels in piggeries. Because most pigs are slaughtered from 6–8 months of age, and commercial farms have high annual turnovers, vaccination is costly and impractical. Furthermore, maternal antibodies render the live-attenuated vaccine ineffective in pigs less than three months old [75], resulting in a narrow period of effective immunity in production pigs. However, in areas where JEV is endemic and pigs live for longer than a few months, pig vaccination may be an effective means of reducing the risk of JE to humans and preventing reproductive losses in sows [93]. With JEV infection in pigs producing limited adverse events, impoverished pig farmers have little economic incentive to immunise swine, and thus the uptake of these vaccines is likely to remain low in developing countries where JE is endemic. 

Removal of pigs from areas of high human population density in JEV endemic regions has proven useful in reducing JE burden. Malaysia responded to a JEV outbreak in 1998–1999 by culling ~50% of swine in the region, which effectively stopped further clinical JEV infections [94]. The reduction in JE incidence in Japan, Taiwan, and Korea may also be partially attributed to removal of piggeries from population centres [2,78]. In the Torres Straits of Australia, pigs were removed from the community on Badu Island and relocated to a piggery ~2.5 km away following an outbreak in 1995 [13,28]. Despite this, JEV was still detected in the community in a subsequent survey [28]. It was found that the piggery was located within the flight range of vector mosquitoes to the community; the presence of viraemic waterbirds nearby the community may also have been an alternative source of mosquito infection. JEV transmission has been maintained in other areas where pig populations are low or where pigs have been removed. For instance, JEV remains endemic in Bangladesh despite the limited number of piggeries due to the diet of the predominantly Muslim population [38,93]. Singapore abolished pig farms in 1992, which reduced JEV infection from 14 cases annually to 3 imported cases from 1991 to 2000 [11]. However, recent surveillance efforts have shown zoonotic transmission of JEV involving wild boars and a range of domestic and wild migratory birds including eagles, egrets, plovers and redshanks [11,95]. Thus, JEV amplification in ardeid birds (egrets, herons) may be important where domestic swine populations are absent, or numbers are low.

### 7.4. Other Hosts

Other domesticated (cattle, horses, buffalo, and goats) and wild (marsupials) animals are considered dead-end-hosts for JEV transmission. These animals may be infected with JEV but develop low viraemia, incapable of transmitting to mosquitoes [25,33,96,97,98,99,100]. In situations where dead-end hosts are present in sufficient numbers and are preferred sources of blood meals by vectors, they may act to dampen or suppress JEV activity by subverting vectors away from amplifying hosts. A recent study conducted in Cambodia observed that JEV may be circulating between multiple hosts including dogs and domestic birds (chickens, ducks), with the presence of JEV neutralizing antibodies detected in a range of animals [101,102]. Most JEV infections in cattle and horses are subclinical. Although bovine JE is rare [103], JE in horses is well recognised and outbreaks in horses parallel those in humans [99]. Vaccination of horses against JEV is mandatory in Hong Kong (China), Malaysia, Japan, and Singapore, which has reduced JEV infection in horses in these areas [104]. Unlike domesticated birds, very little is known about JEV viraemia in newborn calves or foals and whether higher viraemia could occur in these younger animals [103]. However, maternal antibodies may protect calves and foals leading to low JEV infection until they have been weaned. Due to relatively high levels of seropositivity, goats and cattle are considered suitable animals for surveillance purposes. In Sri Lanka, seroprevalence in cattle and goats was found to be a better predictor of the JE infection risk for humans than porcine seroprevalence [105]. This is likely due to the vector feeding preference for these species over pigs.

JEV antibodies have also been detected from a large range of bat species in East Asia [106,107]. Indeed, JEV has been isolated from bats in Japan [108], Taiwan [109], and China [107,110]. Experimental studies have mainly focussed on microbats, in which viremia has been found to last as long as 25–30 days at a level sufficiently high to infect mosquitoes [111]. Transplacental virus transmission has also been demonstrated experimentally, and this may be a mechanism by which JEV can be maintained in nature [112]. Experimental infections of a megabat species (Pteropus alecto) also demonstrated that animals infected by inoculation or via infected mosquitoes developed viraemia capable of infecting recipient mosquitoes [113]. Bats have also been proposed to play a role in the over-wintering of JEV in the northern range of its distribution, via reactivation of viraemia in infected bats following hibernation [111]. Further studies are needed to establish the role of bats in the natural cycle of JEV circulation.

## 8. Possible Impacts of Climate Change

The changing climate, including an increase in global temperature and changes in precipitation and wind patterns, can have a profound effect on the distribution of the viral vectors, reservoir species, amplifying hosts, and even the genotypes of JEV. In addition to increasing temperatures, potential regions where JEV can exist may experience longer and more intense periods of low rainfall or drought as well as periods of higher than usual rainfall and possibly an increase in the frequency and intensity of extreme weather events such as cyclones or hurricanes.

The drivers of JEV ecology and epidemiology are complex, with contributions from hosts, vectors, viral genetics, and human behaviour, and these are influenced by a range of factors including climate [81,114,115,116,117,118,119]. Increases in temperature can have effects on the development time, immature survival, adult survival, mosquito size, blood feeding, and fecundity of *Culex* mosquitoes. There may be changes in vector competence and the extrinsic incubation period [120,121,122]. Fluctuating diurnal temperatures are an important aspect of field conditions, and these may have a greater effect than can be predicted solely from increases in mean temperature [121]. 

Changes in temperature can also have some effect on the dispersal of aquatic birds such as herons, which may be infected with JEV without manifesting clinical signs. Daytime and night-time temperatures can be an important component of the modelling for the distribution of the reservoir species [123].

Viral genotype may vary with climate, with evidence suggesting that GIII and GI-b are temperate genotypes, while G1-a and GII are tropical genotypes [124]. However, further studies are needed to establish these trends, as these associations may be due to bias introduced from sampling and sequence availability. Further studies are needed to determine if genotype-defining substitutions can provide some insights into the molecular mechanisms responsible for this phenomenon [124]. 

Prolonged dry periods can be associated with a reduction in natural breeding sites for mosquitoes and their reservoir species, resulting in reduced transmission of JEV. The reduction in river flows can reduce the supply of irrigation water with a substantial reduction in rice fields contributing to a further reduction in transmission. Amplifying species such as pigs will be constantly replaced. Vaccination of pigs is rare, and when there is minimal transmission, the incentive for human vaccination may also be reduced. This scenario can result in a substantial increase in the susceptible populations of both pigs and humans. When a heavy rain event occurs with resultant flooding, there can be long distance dispersal of JEV by aquatic birds and massive amplification by immunologically naive pigs [119,123], resulting in spillover into the susceptible human population.

Mosquitoes can be dispersed relatively long distances by wind [125,126]. The roles of cyclones and similar weather events in the dispersal of infected Culicoides and bluetongue outbreaks has been well documented [127,128,129,130,131,132], and there is also limited evidence that windblown mosquitoes can introduce JEV into new areas [133]. The increased frequency and intensity of these events with global warming may introduce or enhance the distribution of JEV into areas that presently do not experience outbreaks. Future trends in Asia will almost certainly include an expansion of the endemic and epidemic regions with very distinct seasonal variation [134].

## 9. JEV Vaccination

Several JEV vaccines have been developed and are currently used in some JEV endemic countries. Inactivated vaccines derived from mouse brains or cell culture have proven to have high efficacy throughout Southeast Asia [135,136]. Initial trials in Taiwan, 1965, demonstrated 80% efficacy [136], but improved purification steps led to reports of 80–100% efficacy in adults from India [137], Japan [138], Thailand [139], and USA [140]. However, high production costs (USD 3–5 per dose), poor induction of long-term immunity, multiple dose regimen, and reports of adverse events limited the use of this vaccine [140,141]. Inactivated vaccines have gradually been replaced with a live-attenuated (SA14-14-2; GIII strain) vaccine, which is given in two to three doses during childhood and is inexpensive to make (USD 0.05 per dose) [142]. The live-attenuated vaccine is immunogenic, induces long term immunity, and requires one or two doses in childhood [143]. In China and Nepal, two dose regimens in children led to seroconversion in 97.5% and 98% of recipients, respectively, with high levels of antibody persisting for up to five years following immunisation [144,145]. While initially only available in China, it is now the recommended vaccine for many endemic countries in Asia [146]. A recombinant, live attenuated version of the SA14-14-2 JEV vaccine has also been developed, which has proven safe and effective, with 97% of children seroprotected 5 years after the booster dose [147]. 

However, some issues remain with these live attenuated vaccines. Neutralising antibodies elicited by JEV GIII poorly neutralised GV virus; and mice immunised with the SA14-14-2 were poorly protected against GV challenge [148]. GV virus has emerged in South Korea and Tibet, which suggests potential for JEV GV outbreaks in SA14-14-2 vaccinated individuals due to the poor vaccine efficacy against this genotype [149,150]. The dominance of JEV GI has raised concerns about the efficacy of GIII vaccines. Serum from humans vaccinated with GIII inactivated vaccine was able to cross-neutralize viruses belonging to GI but with reduced levels of neutralisation [151,152]. Similar studies using animal antisera from GIII-vaccinates also showed limited cross-protection to local GI isolates [153,154]. Further studies are therefore needed to determine the efficacy of GIII JEV vaccines.

JE vaccine strategies include campaigns, routine immunisation programs, or a combination of both. The most effective immunisation strategy reported is a one-time campaign that typically targets children (<15 years), followed by incorporation of the vaccine into routine immunisation programs [155]. In Nepal, mass vaccination campaigns (2006–2009) targeted at children (1–15 years) led to 94% coverage of the target population [156]. From 2004–2009, incidence of confirmed JEV following the campaign was 1.3 per 100,000, which was 72% lower than the expected incidence of 4.6 per 100,000 prior to the vaccination campaign [157]. In 2017, a JEV vaccine campaign was initiated in Myanmar with 14 million children (5–15 years) targeted for immunisation followed by implementation of routine immunisation from 2018 [158]. The campaign immunised ~12.6 million children, resulting in nationwide coverage of 92.5% [158]. No data have yet been published for the effect this campaign has had on JEV circulation in Myanmar. A comprehensive study on 14 endemic countries from the period of 2007 to 2021 estimated the outcomes of campaign and routine immunisation programs to result in a decrease of 193,676 JE cases, 43,446 deaths, and 77,470 cases with sequelae, thereby reducing 6,622,923 disability-adjusted life years (DALYs) and saving USD19 million in AME cases [159]. Thus, the implementation of JE vaccine programs in endemic countries are expected to be highly beneficial. 

Vaccination programs are increasingly helping to control JEV, but better surveillance is needed to improve estimates of JEV burden. Underreporting is a key problem for understanding the local and global burden of JE and to help better identify areas at risk of disease. Surveillance guidelines are available from the WHO [160]. However, many JE endemic countries have limited JEV diagnostic capability.

To address this issue, a WHO JEV laboratory network, modelled after polio and measles laboratory networks, has been developed [160,161]. The network aims to build diagnostic and surveillance capacity through training and development. Already, the network has improved surveillance in the Asian region from 75% to 92% of endemic Asian countries now conducting JE surveillance [162]. More than 50% of surveillance programs were at the national level, with other countries conducting subnational or sentinel surveillance only [162]. Despite an overall 60% drop in JE cases from 2011–2015, Nepal and Vietnam had an 11-fold and 2-fold rise in reported JE cases, respectively [162]. Such reports are most likely due to improvements in surveillance practices, coupled with prior inconsistencies in reporting.

## 10. Conclusions

JEV is an ongoing public health threat in many Asian countries. Although safe and effective vaccines are available, vaccination rates remain low in most countries where they are most urgently needed. There is a risk of further expansion of JEV into new regions due to the lack of effective vector control programs, the geographic expansion of human-adapted mosquito species, and the possible impacts of climate change.

## Figures and Tables

**Figure 1 pathogens-10-01534-f001:**
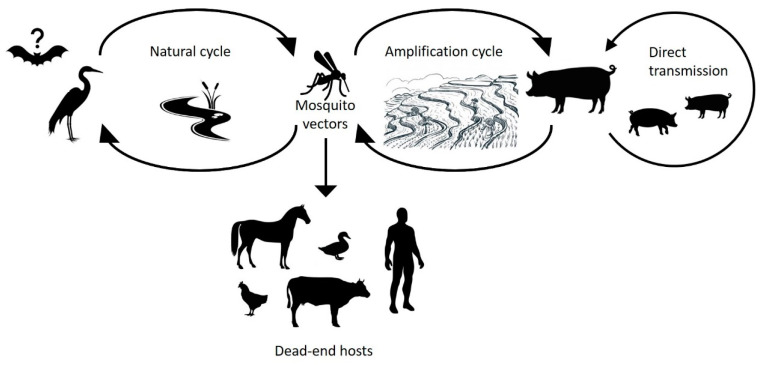
Overview of the Japanese encephalitis virus transmission cycle.

**Figure 2 pathogens-10-01534-f002:**
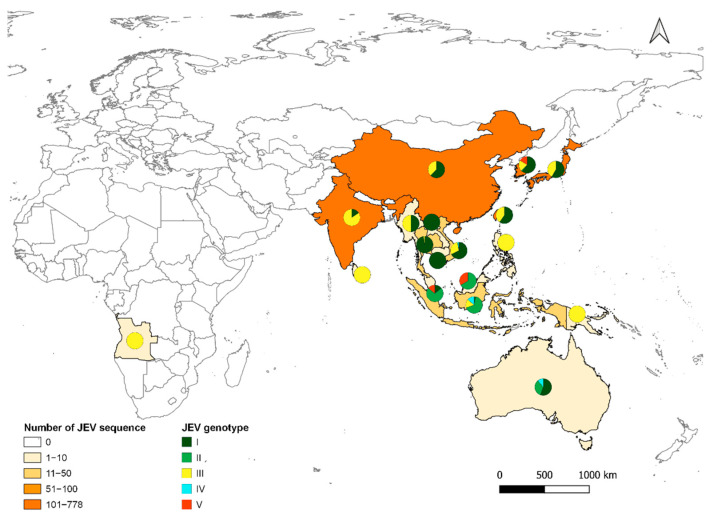
Map of the distribution of JEV genotypes based on available C/prM, envelope, NS, and 5’UTR sequences (*n* = 1840). Strains with different gene sequences (same strain ID) or 100% identity were removed.

## Data Availability

Not applicable.

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
