# Peer review of "The Ecology and Evolution of Japanese Encephalitis Virus"

_pathogens, 2021, doi:10.3390/pathogens10121534_

Round 1

Reviewer 1 Report

This review article from Mulvey et al is a comprehensive summary of the evolution and ecology of Japanese encephalitis virus, with additional brief descriptions of its virology and pathology.  As a whole it is well organized and enjoyable to read, and is especially pertinent given the likely alterations to mosquito populations with global climate change.  I only have a few minor comments to be addressed:

1) The manuscript could use 1-2 more figures in addition to the provided map.  Possibilities include a figure summarizing the spread of JEV among its various hosts and a figure summarizing the impacts of  climate change on JEV spread.

2) On page 3 line 104, the authors should specify what AME stands for.

3) On  page 4 line 137, the authors should specify the means by which GIII was predicted to have a greater selective advantage- computationally, experimentally, etc.  The authors could also indicate whether there's any clear trends for substitutions that are under positive selection.

4) On page 6 line 272, the authors should specify whether long term JEV persistence in pig tonsils was identified with infectious virions or with viral RNA.

5) On page 7 line 323, the  authors could list a few other species in addition to dogs that may act as circulating hosts

6) On page 8 line 333, the authors should clarify why seroprevalence in cattle and goats was thought to be a better predictor of JEV prevalence in local human populations than pig seroprevalence given the more established role of pigs in spreading JEV.

Author Response

Reviewer 1

This review article from Mulvey et al is a comprehensive summary of the evolution and ecology of Japanese encephalitis virus, with additional brief descriptions of its virology and pathology.  As a whole it is well organized and enjoyable to read, and is especially pertinent given the likely alterations to mosquito populations with global climate change.  I only have a few minor comments to be addressed:

1) The manuscript could use 1-2 more figures in addition to the provided map.  Possibilities include a figure summarizing the spread of JEV among its various hosts and a figure summarizing the impacts of climate change on JEV spread.

An extra figure has been added that depicts the transmission cycles of JEV (Figure 1).

The possible impacts of climate change on the distribution and risk of JEV is an area that requires further analysis as the associated factors are very complicated. Further modelling of different climate scenarios may provide some clarity. Due to these complexities, and the paucity of clear data, we believe that a figure depicting the potential impacts of climate change on JEV distribution is beyond the scope of the present review article.

2) On page 3 line 104, the authors should specify what AME stands for.

AME was defined on the first mention in the article (Page 2, Line 71).

3) On page 4 line 137, the authors should specify the means by which GIII was predicted to have a greater selective advantage- computationally, experimentally, etc.  The authors could also indicate whether there's any clear trends for substitutions that are under positive selection.

We have added further information to a part of the paragraph indicated where we make it clearer that we were referring to in silico experiments. In general, we have clearly stated throughout the indicated paragraph where studies were experimental or computational; and we have also indicated where we have discussed whether mutations were under positive selection. We have highlighted these points below as further evidence:

In an attempt to identify genetic determinants driving GI-b/GIII lineage replacement, GI-b and GIII genome and E protein sequences from Asian strains were analysed in silico to investigate amino acid mutations, positive selection and host range [43]. Both lineages had low ratios of non-synonymous to synonymous substitutions (dN/dS) with a small number of individual sites under positive selection [43]. Historically, identification of positive selection in the genome of an emerging lineage would suggest a selective advantage. Surprisingly, GIII was shown to have a greater selective advantage because it was predicted to be shaped by positive selection, while GI was predicted to be neutral [43]. The lower diversity of GI strains was associated with narrower vector range than GIII strains suggesting enhanced transmission or more efficient replication cycle between Culex sp. and pigs. This theory was supported by in vitro experiments that observed GI isolates with significantly higher infectivity titres in mosquito cell lines (C6/36) compared to GIII isolates [56]. GI isolates had more than an 11-fold increase in titres compared to GIII isolates, which suggested GI had better replicative fitness in mosquitoes and therefore represent a greater risk of transmission [56]. In another study, oral challenge of Cx. quinquefasciatus mosquitoes with either GI-b or GIII isolates resulted in no significant differences in virus ingestion, dissemination, or transmission rates in these mosquitoes [55]. However, a higher infection rate was found for the GIII isolate tested compared to the GI isolates [55], further confounding interpretations of the basis for the dominance of GI. Therefore, the biological mechanism underlying the GI replacement of GIII throughout Asia is not yet clear. Future in vivo challenge studies with the primary vector in Southeast Asia, Cx. tritaeniorhynchus, may be revealing.

4) On page 6 line 272, the authors should specify whether long term JEV persistence in pig tonsils was identified with infectious virions or with viral RNA.

The sentence was amended as suggested:

“JEV RNA and live virus persisted in pig tonsils for six weeks which was likely the primary site of replication and transmission [93].”

5) On page 7 line 323, the authors could list a few other species in addition to dogs that may act as circulating hosts

The sentence was amended as suggested:

A recent study conducted in Cambodia observed that JEV may be circulating between multiple hosts including dogs and domestic birds (chickens, ducks), with the presence of JEV neutralizing antibodies detected in a range of animals [106,107].

6) On page 8 line 333, the authors should clarify why seroprevalence in cattle and goats was thought to be a better predictor of JEV prevalence in local human populations than pig seroprevalence given the more established role of pigs in spreading JEV.

An extra sentence was added to give a possible explanation for this observation:

In Sri Lanka, seroprevalence in cattle and goats was found to be a better predictor of the JE infection risk for humans than porcine seroprevalence [110]. This is likely due to the vector feeding preference for these species over pigs.

Reviewer 2 Report

Summary

In this review, the authors assess the ecology and evolution of Japanese Encephalitis Virus (JEV), which is known to cause encephalitis in 1% of infected human population with associated sequalae in survivors.  Main vector of JEV transmission are mosquitoes, but pigs act as amplifying host and ardeid water birds are also reservoirs. Mosquito-borne diseases are important to study especially in tropical and temperate climate where they cause severe illness with high mortality. Authors look into the epidemiology, genetic evolution and impact of climate change and raise concerns on a need to increase surveillance and testing to limit the spread. The authors did a good job in expanding on key hosts responsible for the spread, impact of economic development on JE and how climate change can increase JE risk areas.

Comments:

  • A figure depicting virus life cycle along with structure of virus with labeled proteins, will be beneficial. May be even a figure with the transcript showing genome organization.
  • It would be nice if the author can comment on the host factors involved in JEV entry and how that affects host tropism and potentially spread.
  • The authors have proposed possible impact of climate change on the spread and transmission of JEV. It would be good if the authors can also comment on other means of potential spread, for example through air travel.
  • Line 112-113: “However, all JEV strains belong to a single serotype because neutralising antibodies are cross-reactive between genotypes”. Reference is missing here.

Author Response

Reviewer 2

In this review, the authors assess the ecology and evolution of Japanese Encephalitis Virus (JEV), which is known to cause encephalitis in 1% of infected human population with associated sequalae in survivors.  Main vector of JEV transmission are mosquitoes, but pigs act as amplifying host and ardeid water birds are also reservoirs. Mosquito-borne diseases are important to study especially in tropical and temperate climate where they cause severe illness with high mortality. Authors look into the epidemiology, genetic evolution and impact of climate change and raise concerns on a need to increase surveillance and testing to limit the spread. The authors did a good job in expanding on key hosts responsible for the spread, impact of economic development on JE and how climate change can increase JE risk areas.

1) A figure depicting virus life cycle along with structure of virus with labeled proteins, will be beneficial. May be even a figure with the transcript showing genome organization.

A figure depicting the lifecycle of JEV has been added, as also recommended by reviewer 1 (Figure 1). We do not agree with the need to add a figure showing the genome organization or transcription factors as this has not been discussed in this review article.

2) It would be nice if the author can comment on the host factors involved in JEV entry and how that affects host tropism and potentially spread.

A paragraph has been added discussed this topic, as suggested by the reviewer:

Host cell tropism is likely determined by JEV attachment and entry into host cells. Initial interaction of JEV to the host cell is thought to be through non-specific attachment of the virion, followed by highly specific binding of the envelope (E) protein to an unknown receptor [17]. Although much is understood about the viral processes involved in viral attachment and entry, little is known about the cellular aspects of this important process. JEV is able to infect and replicate in a range of cell types from different animal species, including mammals, birds and insects. As such, there are probably numerous cell factors involved in viral attachment and entry. Further elucidation of the viral-host interactions is an important area of research for possible therapeutics against JEV infection [17].

3) The authors have proposed possible impact of climate change on the spread and transmission of JEV. It would be good if the authors can also comment on other means of potential spread, for example through air travel.

Air travel is not expected to contribute to the spread of JEV as humans are a deadend host.

4) Line 112-113: “However, all JEV strains belong to a single serotype because neutralising antibodies are cross-reactive between genotypes”. Reference is missing here.

A reference was added as suggested:

However, all JEV strains belong to a single serotype because neutralising antibodies are cross-reactive between genotypes [16].